# OpenReview forum: "$\tau^2$-bench: : Evaluating Conversational Agents in a Dual-Control Environment"
_ICLR.cc/2026/Conference — ICLR 2026 Conference Desk Rejected Submission_

### Official Review · Reviewer_DvCS · 2025-10-28

**Soundness:** 3
**Presentation:** 3
**Contribution:** 3
**Rating:** 8
**Confidence:** 5

**Summary:**

This is an extension of Tao-Bench with an additional new domain of customer support. For that additional domain, they use sophisticated DEC-POMDP approach for establishing agent and user simulator dialogues.

**Strengths:**

The paper reads like an incremental work, like adding a third domain to well known Tao-Bench. This is an under-sell. The authors should emphasize the decentralized multi-agent POMDP based platform for establishing user simulator and agent conversations. The content should also provide more details on how these POMDP models are trained.

**Weaknesses:**

One big weakness of this benchmark, and same for Tao-Bench, is that the dialogues only follow some golden path towards task completion. In real life the users are not that collaborative, and there are a bunch of back and forth with the customer service reps. This enables differentiating real life agents being robust to such error recovery cases, and user characteristics. Like users sometimes say "I do not understand what you want me to do" or "I hate this phone, nothing works", and good agents are supposed to handle such cases in a more realistic benchmark. The second concern is that the business rules, defining the dialogue workflow is trivial. There are no such policy actions, as in dialogue managers to handle. For example, one rule can say, if the phone model is X do not suggest Y unless the OS is at least version Z. And one can have thousands of such policy rules. Most real world customer service representatives deal with such policy guidelines. Maybe these can be mentioned as future work.

**Questions:**

I assume POMDP is not used to train the models (since proprietary models are used), but to train a collaboration model. That part is not clear. What type of a model is trained using POMDP and how exactly is it used for prompting closed models.

---

> ### Author Response · Authors · 2025-11-25
>
> We thank the reviewer for the positive assessment and for highlighting the value of our Dec-POMDP-based platform. We would like to clarify our methodology and address your remaining concerns with the following responses.
>
> **1\. Clarification on Dec-POMDP and Model Training**
>
> > The authors should emphasize the decentralized multi-agent POMDP based platform... provide more details on how these POMDP models are trained. ... I assume POMDP is not used to train the models ... but to train a collaboration model.
>
> We clarify that in the current scope of $\\tau^2$-bench, the Dec-POMDP is used strictly as a **formalism to define the environment**, rather than to train a model.
>
> * **Environment Definition:** We use the Dec-POMDP tuple to strictly define the interaction dynamics, state transitions, and observation spaces. This enables the standardized evaluation of off-the-shelf LLMs (e.g., GPT-4, Claude-3.7) in a dual-control setting without requiring task-specific fine-tuning.
> * **Enabling Future RL Research:** We agree with the reviewer’s intuition that this platform is well-suited for training. By formalizing the problem as a Dec-POMDP, $\\tau^2$-bench provides the necessary infrastructure for future work to apply Reinforcement Learning (RL) or Multi-Agent RL to train explicit collaboration policies. We will emphasize this potential in the revision.
>
> **2\. User Realism and the "Golden Path"**
>
> > One big weakness... is that the dialogues only follow some golden path towards task completion. In real life the users are not that collaborative...
>
> We agree that handling non-collaborative users is critical. $\\tau^2$-bench moves beyond the "golden path" by explicitly modeling **User Personas** to test agent robustness.
>
> * **Hard Personas:** We introduce a "Hard" persona representing users with low technical knowledge (e.g., instructions state: "Step-by-step instructions often confuse you"). This directly models the confusion scenarios mentioned by the reviewer.
> * **Robustness Evaluation:** Our results demonstrate that this design successfully differentiates agent robustness: performance drops significantly for the "Hard" persona compared to the "Easy" persona across all models (e.g., GPT-4 drops from \~0.40 pass rate to \~0.36), confirming that agents are tested on error recovery.
> * **Future Work:** We leave further extension that explores this aspect as future work.
>
> **3\. Complexity of Rules vs. Interaction Dynamics**
>
> > The second concern is that the business rules... is trivial. ... one can have thousands of such policy rules.
>
> While individual rules in the Telecom domain are designed to be intuitive, the benchmark's difficulty stems from **coordination friction** rather than static logical complexity.
>
> * **Coordination is the Bottleneck:** Our ablation studies show a substantial performance drop (\~20% for GPT-4) when moving from a single-agent ("No-User") setting to the dual-control ("Default") setting. This confirms that the primary challenge is the *information asymmetry* and the need to *guide* an active user, independent of rule complexity.
> * **Complementarity:** We included the original $\\tau$-bench domains (Retail and Airline) specifically to cover scenarios with complex policy constraints. The Telecom domain was designed to isolate and stress-test the novel **dual-control** capabilities absent in prior benchmarks.
> * **Future Work:** We agree that scaling to "thousands of rules" while maintaining dual-control interaction is a valuable direction for future stress-testing and will note this in the discussion.

---

### Official Review · Reviewer_CRMZ · 2025-11-01

**Soundness:** 3
**Presentation:** 3
**Contribution:** 4
**Rating:** 8
**Confidence:** 4

**Summary:**

The paper introduces τ²-Bench, an extension of τ-Bench that evaluates conversational AI agents in dual-control environments where both the agent and the user possess tool-using capabilities. It defines a Dec-POMDP-based setup allowing both sides to act on a shared world state, implemented in a telecom domain with compositional task generation and a tightly coupled user simulator. Experiments across multiple models (GPT-4.1, o4-mini, Claude-3.7-Sonnet) show marked performance degradation when moving from single-control to dual-control settings, isolating reasoning versus coordination failures. The benchmark also provides diagnostic insights into user simulation reliability and introduces structured task creation pipelines ensuring domain coverage and controlled complexity.

**Strengths:**

The paper makes a clear and well-motivated advancement in benchmarking agentic systems by introducing the dual-control paradigm, a crucial step toward realistic evaluation of human-AI collaboration. The Dec-POMDP formalization provides a principled theoretical grounding and distinguishes τ²-Bench from purely empirical benchmarks. The telecom domain design, complete with compositional task generation and programmatic verification, adds rigor and reproducibility. The ablation studies are also insightful and the comparison of dual-degree v/s single degree User Simulation is also very insightful and could be helpful for other domains as well where LLMs are utilised to simulate real world users.

**Weaknesses:**

While the benchmark is thoughtfully designed, the paper could better contextualize why the dual-control setting matters beyond the telecom example. Also, the evaluation focuses primarily on accuracy (pass^k metrics), with little analysis of qualitative coordination behaviors, there has been a recent surge in Communication protocols for multi agents systems such as google's Agent-to-Agent, the current metrics primarily focuses on evaluating the final outpu, extending the metrics to incorporate more qualitative aspects can further enhance the usability of the benchmark into properly evaluating more complex aspects of multi agent systems.

**Questions:**

* How do existing AI or agentic frameworks (for example, tool-use agents or dialogue planners) align with the dual-control abstractions introduced here? Which components are novel in practice rather than merely reformulated theoretically?

* How do the theoretical coordination challenges observed in τ²-Bench correspond to practical system limitations seen in real multi-agent or user-in-the-loop deployments?

* Could controlled experiments involving minimal cross-domain extensions (for example, retail dual-control) or user simulator variants demonstrate the benchmark’s generality and stability more concretely?

Also refer the weaknesses section above

---

> ### Author Response · Authors · 2025-11-25
>
> We thank the reviewer for recognizing the value of the dual-control paradigm and Dec-POMDP formalization. We address your specific questions and suggestions below.
>
> **1\. Contextualizing Dual-Control Beyond Telecom**
>
> > the paper could better contextualize why the dual-control setting matters beyond the telecom example.
>
> The telecom domain is representative of a much broader class of real-world problems.
>
> * **Ubiquity of Technical Support:** Technical support spans industries (e.g., software troubleshooting, IT helpdesk, IoT device setup). In these scenarios, the agent cannot directly manipulate the user's device or local software state and must guide the user to perform actions.
> * **Information Retrieval:** Dual-control extends beyond state changes to information retrieval. Any scenario where an agent guides a user to retrieve private information via local tools (e.g., checking a calendar, searching emails, verifying bank details) falls under the dual-control formulation.
>
> **2\. Qualitative Analysis vs. Programmatic Verification**
>
> > the evaluation focuses primarily on accuracy (pass^k metrics), with little analysis of qualitative coordination behaviors ... extending the metrics to incorporate more qualitative aspects can further enhance the usability
>
> We agree qualitative analysis is valuable, but prioritized **programmatic verifiability** to ensure reliability.
>
> * **Reliability vs. Nuance:** Automated qualitative metrics for coordination are challenging and often require human judgment or model-based evaluation, introducing noise. We maintain a strict, verifiable "pass/fail" standard to establish a reliable baseline.
> * **Quantitative Proxies:** Instead of subjective metrics, we derive insights through quantitative **ablation studies**. By measuring performance across **Personas** (Easy vs. Hard), **Policy Documentation** (Original vs. Workflow), and **Task Complexity** (number of sub-tasks), we indirectly quantify aspects like robustness and coordination efficiency.
>
> **3\. Alignment with Existing Frameworks and Novelty**
>
> > How do existing AI or agentic frameworks ... align with the dual-control abstractions introduced here? Which components are novel in practice rather than merely reformulated theoretically?
>
> * **Alignment:** Existing agentic frameworks (e.g., LangChain, AutoGen) focus on constructing the *Agent* instance. $\\tau^2$-bench is agnostic to how the agent is built; it provides the **Environment**. Any framework capable of tool use and dialogue can be instantiated as the Agent (or User) within our domain.
> * **Novelty:** The primary practical novelty is the **Dec-POMDP environment** for text-based technical support. While Dec-POMDP is an established concept, applying it to formulate a dual-control grounded environment—where specific tools define the action space for both sides—moves benchmarking beyond intent classification or single-agent execution.
>
> **4\. Real-World System Limitations**
>
> > How do the theoretical coordination challenges observed in $\\tau^2$-Bench correspond to practical system limitations ...?
>
> The coordination overhead measured in our benchmark maps to two practical limitations in real-world user-in-the-loop deployments:
>
> * **Information Asymmetry (User fails to retrieve info):** Users often lack the context to proactively mention key information the agent needs (e.g., error indicators or device status). $\\tau^2$-bench captures this by modeling user state as partially observable, forcing multi-turn information elicitation rather than assuming perfect knowledge.
> * **Execution Friction (User fails to execute actions):** Users frequently fail to execute instructions precisely due to confusion or lack of proficiency. Our dual-control setup simulates this gap, where the agent must not only generate a correct plan but successfully guide the user to perform state-changing actions (e.g., toggling settings), mirroring remote troubleshooting friction.
>
> **5\. Cross-Domain Extensions**
>
> > Could controlled experiments involving minimal cross-domain extensions (for example, retail dual-control) ... demonstrate the benchmark’s generality ...?
>
> We appreciate this suggestion.
>
> * **Nature of Existing Domains:** The existing Retail and Airline domains in $\\tau$-bench were inherently designed for single-control settings (i.e., the agent performs actions *on behalf* of the user, such as booking a ticket). Forcing these into a dual-control setting does not fundamentally alter the task nature in a meaningful way.
> * **Future Work:** We believe the dual-control paradigm is best demonstrated in domains where collaboration is essential (like Technical Support). We prioritize expanding to new, inherently collaborative domains in future work rather than retrofitting domains designed for single-agent delegation.

---

### Official Review · Reviewer_BQA3 · 2025-11-04

**Soundness:** 3
**Presentation:** 3
**Contribution:** 3
**Rating:** 8
**Confidence:** 3

**Summary:**

This paper introduces τ²-Bench, a novel benchmark for evaluating conversational agents in dual-control environments, where both the user and the agent can perform actions to affect the shared world state. Unlike prior single-control setups (τ-Bench), τ²-Bench models realistic collaborative tasks, such as telecom troubleshooting, using a Decentralized Partially Observable Markov Decision Process (Dec-POMDP) formulation. The benchmark features:

- A compositional task generator that assembles complex, multi-step tasks from atomic subtasks.
- A dual-control user simulator that interacts consistently and realistically with the agent.
- An evaluation methodology that separates reasoning from communication/coordination via ablation settings (Default, No-User, Oracle Plan).

Experiments on telecom, retail, and airline domains reveal that large LLMs (GPT-4.1, Claude-3.7-Sonnet, etc.) experience a significant performance drop (~20%) when coordinating with users compared to acting alone. The benchmark also demonstrates improved simulator reliability and fine-grained diagnostic capabilities compared to prior frameworks.

**Strengths:**

1. Dual-Control Realism – Unlike τ-Bench, which assumes single-agent control, τ²-Bench models dual-control settings where both user and agent can act. This reflects realistic collaborative tasks (e.g., troubleshooting or co-working) and exposes coordination challenges absent in prior benchmarks.

2. Reduced Simulator Errors and Improved Reliability – τ²-Bench’s affordance-based simulator design cuts total error rates from ~40–47% in τ-Bench to ~16% (6% critical), making evaluations far more stable and trustworthy.

3. Compositional Task Generator for Greater Scalability and Control – τ²-Bench replaces τ-Bench’s hand-curated tasks with a programmatic, compositional framework that builds tasks from atomic subtasks (`init`, `solve`, `assert`). This ensures *logical correctness*, *diversity*, and *reproducibility*, while enabling fine-grained control over task complexity and automatic generation of thousands of domain-consistent scenarios, offering far greater scalability and flexibility than τ-Bench’s static design.

4. Fine-Grained Analysis – τ²-Bench separates reasoning (no-user mode) from communication/coordination (dual-control mode), which allows to pinpoint failure sources. τ-Bench conflated these dimensions, making it difficult to isolate why an agent failed.

5. Quantitative and Qualitative results – τ²-Bench provides detailed ablations and performance decompositions (Default, No-User, Oracle Plan), revealing that LLMs drop ~20% in success rate when coordinating with users, previously not measurable under τ-Bench’s single-agent regime.

**Weaknesses:**

1. **Limited Modularity and Domain Portability** – Although τ²-Bench generalizes τ-Bench conceptually, creating new task domains still requires rebuilding much of the system, including domain-specific tools, task generators, and simulators. This limits modularity and makes scaling to new environments labor-intensive and dependent on expert curation.

2. **Lack of Human-in-the-Loop Evaluation** – The benchmark relies entirely on simulated users and agents, with no human studies to validate realism or real-world transfer. Incorporating human participants, either as users or agents, would provide stronger evidence of practical effectiveness and ecological validity.

**Questions:**

1. **How does τ²-Bench fundamentally differ from τ-Bench beyond the dual-control setting?**
For instance, are there changes in task semantics, evaluation metrics, or simulator-agent interaction logic that might independently contribute to the observed performance drop?

2. **How sensitive is τ²-Bench performance to the choice of agent prompting or policy design?**
Since different LLMs and prompting strategies can give very different communication behaviors, it would be useful to understand whether τ²-Bench exposes consistent weaknesses across models or is heavily policy-dependent.

3. **How transferable are policies across domains?**
If a model performs well in τ²-Bench’s telecom setting, does that translate to improved coordination in τ-Bench’s retail or airline domains?

---

> ### Author Response · Authors · 2025-11-25
>
> We thank the reviewer for recognizing the realism of our dual-control setup and the improved simulator reliability. We address your questions below.
>
> **1\. Limited Modularity and Domain Portability**
>
> > ...creating new task domains still requires rebuilding much of the system... This limits modularity and makes scaling to new environments labor-intensive and dependent on expert curation.
>
> Domain curation follows a structured pipeline involving AI generation and human verification.
>
> * **AI-Assisted Pipeline:** Domain materials (PRD, database schemas, tool implementations) are largely generated by AI models. We maintain a **Human-in-the-loop** approach (Stages 1, 2, 5, see Section 3.2) for quality control.
> * **Future Automation:** This process will become increasingly automated with stronger AI models and refined pipelines, reducing the entry barrier for new domains.
>
> We have updated the paper to include a comparison of curation processes between τ-Bench and τ2-Bench (Table 2), highlighting the path towards automation.
>
> **2\. Lack of Human-in-the-Loop Evaluation**
>
> > The benchmark relies entirely on simulated users... Incorporating human participants... would provide stronger evidence of practical effectiveness and ecological validity.
>
> We agree that human evaluation is the ultimate test for ecological validity. However, we prioritized a **controllable approach** to ensure **reproducibility**.
>
> * **Standardization:** $\tau^2$-bench aims to serve as a standardized benchmark. Human evaluation introduces significant variance, making reproducibility difficult. $\tau^2$-bench is not a replacement for real-world testing, but probes specific agent abilities.
> * **Controlled Proxy:** Our User Simulator uses affordance constraints to minimize hallucinations. Even with this controlled, cooperative simulator, we observe ~20% performance drop compared to single-agent settings. Since real users are less predictable, the actual gap would likely be larger.
>
> **Q1. Distinction from τ-Bench**
>
> > How does $\tau^2$-Bench fundamentally differ... are there changes in task semantics... that might independently contribute to the observed performance drop?
>
> τ²-Bench introduces structural extensions—**but none alter the task semantics used in our performance comparison**.
>
> **What changed in τ²-Bench (not sources of the drop):**
>
> * **Structured user task definitions:** explicit known/unknown information, main intent vs. instructions, and attachable personas. These enrich user modeling but do not change tasks or tool requirements.
>
> * **More expressive verification:** besides DB-state matching, τ²-Bench adds **environment assertions** (e.g., "WiFi restored") for both agent and user actions. Success conditions match τ-Bench equivalents, so evaluation criteria are unchanged.
>
> * **Unified reward:** rewards now integrate both agent and user actions
>
> **Why these do not explain the performance drop:**
> Our ablation isolates user agency by running **the same tasks** and **correctness checks** in two modes:
>
> 1. **No-User:** agent controls all tools (τ-Bench–equivalent).
> 2. **Dual-Control:** agent and user share control.
>
> Models exhibit an **18–25% drop** from No-User → Dual-Control. Because task semantics, tools, and metrics are identical, the decline is attributable to **coordination and communication challenges**, not τ²-Bench changes.
>
> **Additional benefits:** τ²-Bench makes it easy to vary personas, toggle No-User/Dual-Control, and modify domain-policy formats for controlled ablations.
>
> **Q2. Sensitivity to Prompting and Policy**
>
> > How sensitive is $\tau^2$-Bench performance to the choice of agent prompting or policy design? ... exposes consistent weaknesses across models...?
>
> The observed weaknesses are consistent, though specific performance is sensitive to policy format.
>
> * **Consistent Drop:** We observe significant performance drops across diverse models (GPT-4, o4-mini, Claude-3.7). This universality indicates current LLMs consistently struggle with the reasoning-communication gap in dual-control environments.
> * **Policy Sensitivity:** We compared "Original Policy" vs. "Workflow-based Policy." As shown in Figure 4, while the Workflow policy improved performance, the coordination gap remained, validating that the benchmark captures the core difficulty.
>
> **Q3. Transferability Across Domains**
>
> > How transferable are policies across domains? ... does that translate to improved coordination in $\tau$-Bench's retail or airline domains?
>
> As shown in Figure 3, similar trends are observed across domains and models. However, this does not imply a better model on telecom is also better on other domains. We hypothesize that dual-control skills are a superset of single-control skills; the telecom domain tests communication, while retail and airline domains test policy adherence.

---

> ### Author Response · Authors · 2025-11-26
>
> We also paste the new Table 2 here for easier reading.
>
> | Phase | τ-Bench (AI Usage) | τ-Bench (Human Effort) | τ2-Bench (AI Usage) | τ2-Bench (Human Effort) |
> | :---- | :---- | :---- | :---- | :---- |
> | **Agent Schema & Tools**  | **Low** \- Limited assistance | **High** \- Manually hand-crafted | **High** \- Generates PRD, schema, and code | **Medium** \- PRD, schema, and code refinement |
> | **User Schema & Tools** | **N/A** | **N/A** | **High** \- Generates PRD, schema and code | **Medium** \- PRD, schema, and code refinement |
> | **Task Creation** | **Low** \- No automation | **High** \- Manually writes task case with unique solution | **None** \- Programmatic verifiable generation via atomic subtasks | **Low** \- Defines atomic subtasks and logic |
> | **Agent Policy** | **High** \- Generates domain-specific policies | **Medium** \- Manually design business rules | **High** \- Generates domain-specific policies and user manuals | **Low** \- Refines logic and details |
> | **Manual Refinement** | **None** | **High** \- debug cases | **Low** \- help with code and document refinements | **High** \- Joint refinement of all materials |

---

### Official Review · Reviewer_rbPG · 2025-11-10

**Soundness:** 3
**Presentation:** 4
**Contribution:** 3
**Rating:** 8
**Confidence:** 3

**Summary:**

τ²-Bench introduces a novel benchmark for evaluating conversational agents in a dual-control setting, meaning both the AI agent and the user can perform actions that change the environment state.Unlike prior benchmarks such as τ-Bench,which modeled multi-turn tool use with a passive user. τ²-Bench addresses a missing realism: the user is an active participant. It simulates scenarios (in a telecom technical support domain) where the user must collaborate with the agent by also using tools to modify the shared world state. It interoduce a compositional task generator, such that tasks are built from atomic subtasks (e.g., connectivity issues, account errors), supporting scalable, diverse, and verifiable evaluations. The benchmark also supports ablation settings (e.g., Oracle Plan, No-User) that help isolate reasoning errors from communication failures, offering insight into agent shortcomings in coordination-heavy tasks.

**Strengths:**

1. Novel Dual-Control Paradigm: The benchmark targets a unique challenge not covered by prior agent benchmarks – the need for an AI agent to collaborate with an active user. This dual-control setup (modeled as a Dec-POMDP) is highly realistic for many applications (e.g., troubleshooting, collaborative tasks) and exposes failure modes (coordination, communication)
2. Comprehensive and Rigorous Benchmark Design: The paper introduces a well-thought-out domain (telecom support) with a structured approach to generate tasks of varying complexity.
3. Strong Experimental Analysis: The evaluation of agent performance is very thorough. The authors perform ablations that isolate different challenges

**Weaknesses:**

1. Limited Domain Scope: A notable limitation is that τ²-Bench’s dual-control evaluation is conducted in only a single domain (telecom support). While this domain is well-chosen and convincingly complex, the benchmark would be stronger if the approach was demonstrated on multiple domains or scenarios.
2. Assumptions in User Simulator Behavior: Another minor weakness is the limited user behavior patterns considered. The user simulator is designed to be cooperative (albeit with possible benign mistakes) and has a fixed set of tools. In reality, users might deviate from instructions, provide irrelevant information, or have varying degrees of proactiveness

**Questions:**

How easily can the τ²-Bench framework be extended to new domains beyond telecom? The current implementation required manual curation of schemas, tools, and policies. Do the authors envision automating more of this process?
Given the Dec-POMDP formulation, one could view the problem as a collaborative multi-agent game between the assistant and the user. Have authors tried applying multi-agent RL or self-play techniques in this context?

---

> ### Author Response · Authors · 2025-11-25
>
> We thank the reviewer for the positive assessment, recognizing the novelty of our dual-control paradigm and the rigorous benchmark design. We address your specific questions and concerns below.
>
> **1\. Limited Domain Scope**
>
> > A notable limitation is that $\\tau^2$-Bench’s dual-control evaluation is conducted in only a single domain (telecom support). ... the benchmark would be stronger if the approach was demonstrated on multiple domains...
>
> Our primary goal is to introduce a scalable framework while also extending the types of domains and tasks that can be implemented in this benchmark. Specifically, our dual-control design enables new types of tasks that were previously not possible, including technical support. With Telecom, we provide one in-depth study of the dual-control framework. This was a deliberate design choice to **prioritize depth and reliability over breadth** in this work.
>
> * **Reliability:** As noted in the paper, existing single-control domains suffer from high user simulator error rates. By adopting affordance-based constraints in the Telecom domain, we ensured a more reliable user simulator, establishing a necessary "Gold Standard" for dual-control environments before scaling.
> * **Verifiability via Composition:** Within this domain, the **Compositional Task Generator** allows us to construct thousands of diverse, verifiable tasks from atomic subtasks. This ensures high reliability compared to hand-curated datasets.
> * **Variations within Domain:** We utilize this depth to study granular variations within the domain, such as different **User Personas** (Easy vs. Hard), to ensure robust evaluation across different interaction dynamics.
> * **Extensibility:** The domain curation method used here is designed to be domain-agnostic and all pipelines can be extended to new domains, as detailed below.
>
> **2\. Extensibility to New Domains**
>
> > How easily can the $\\tau^2$-Bench framework be extended to new domains beyond telecom? ... Do the authors envision automating more of this process?
>
> The curation of new domains follows a structured pipeline involving both AI generation and human verification.
>
> * **AI-Assisted Pipeline:** A large portion of the domain materials (PRD, database
> * schemas, tool implementations) are generated by AI models following our defined pipeline. Currently, we maintain a **Human-in-the-loop** approach (Stages 1, 2, 5, see Section 3.2 DOMAIN AND TASK CREATION) to ensure quality control over the materials.
> * **Future Automation:** We anticipate this process will become increasingly automated with the progress of stronger AI models and refined pipelines, further reducing the entry barrier for creating new domains.
>
> We have updated the paper to include a table comparison of data curation processes between τ \-Bench and τ2-Bench on AI usage and human efforts, highlighting the path towards automation (Table 2).
>
> **3\. User Simulator Behavior and Realism**
>
> > The user simulator is designed to be cooperative... In reality, users might deviate from instructions, provide irrelevant information, or have varying degrees of proactiveness
>
> We appreciate this suggestion.
>
> * **User Personas:** Our study of different **Personas** (Line 463 and Figure 6 \[right\]) aligns with this direction. For instance, our "Hard" persona models users who are less tech-savvy ("flustered quickly") or provide limited information ("only share information when specifically asked").
> * **Cooperative Nature:** We note that the technical support domain is inherently cooperative (the user wants their issue resolved).
> * **Future Work:** While we currently model friction through personas, we leave fully non-cooperative or adversarial user simulation as a direction for future work.
>
> **4\. Multi-Agent RL (MARL) and Self-Play**
>
> > Given the Dec-POMDP formulation... Have authors tried applying multi-agent RL or self-play techniques in this context?
>
> * **Current Scope:** We have not yet applied RL in this specific work, as our primary goal was to benchmark **off-the-shelf LLMs** (e.g., GPT-4, Claude) to understand their native capabilities without fine-tuning.
> * **Enabling Future Work:** The Dec-POMDP formulation facilitates the training of LLM Agents. $\\tau^2$-bench provides the necessary state/action/reward formalism to serve as a **Gym environment** for future research into training models under various settings, including multi-agent RL and self-play.

---

> ### Author Response · Authors · 2025-11-26
>
> We also paste the new Table 2 here for easier reading.
>
> | Phase | τ-Bench (AI Usage) | τ-Bench (Human Effort) | τ2-Bench (AI Usage) | τ2-Bench (Human Effort) |
> | :---- | :---- | :---- | :---- | :---- |
> | **Agent Schema & Tools**  | **Low** \- Limited assistance | **High** \- Manually hand-crafted | **High** \- Generates PRD, schema, and code | **Medium** \- PRD, schema, and code refinement |
> | **User Schema & Tools** | **N/A** | **N/A** | **High** \- Generates PRD, schema and code | **Medium** \- PRD, schema, and code refinement |
> | **Task Creation** | **Low** \- No automation | **High** \- Manually writes task case with unique solution | **None** \- Programmatic verifiable generation via atomic subtasks | **Low** \- Defines atomic subtasks and logic |
> | **Agent Policy** | **High** \- Generates domain-specific policies | **Medium** \- Manually design business rules | **High** \- Generates domain-specific policies and user manuals | **Low** \- Refines logic and details |
> | **Manual Refinement** | **None** | **High** \- debug cases | **Low** \- help with code and document refinements | **High** \- Joint refinement of all materials |

---

### Note · Program_Chairs · 2026-01-17
**Submission Desk Rejected by Program Chairs**

The following references in this submission do not refer to real documents and/or have major errors in bibliographic information:

 Jost Schatzmann, Daniel Jurafsky, Michael Galley, and David Trevillian. Evaluating agenda-based user simulation for reinforcement learning of dialogue management. In Speech Communication, volume 47, pp. 95-121, 2007.